# Computer-Aided Identification of Kinase-Targeted Small Molecules for Cancer: A Review on AKT Protein

**DOI:** 10.3390/ph16070993

**Published:** 2023-07-11

**Authors:** Erika Primavera, Deborah Palazzotti, Maria Letizia Barreca, Andrea Astolfi

**Affiliations:** Department of Pharmaceutical Sciences, “Department of Excellence 2018–2022”, University of Perugia, 06123 Perugia, Italy; erika.primavera@studenti.unipg.it (E.P.); deborah.palazzotti@studenti.unipg.it (D.P.); maria.barreca@unipg.it (M.L.B.)

**Keywords:** cancer, AKT, computer-aided drug discovery, virtual screening, docking, pharmacophore, kinase inhibitors, machine learning, QSAR

## Abstract

AKT (also known as PKB) is a serine/threonine kinase that plays a pivotal regulatory role in the PI3K/AKT/mTOR signaling pathway. Dysregulation of AKT activity, especially its hyperactivation, is closely associated with the development of various human cancers and resistance to chemotherapy. Over the years, a wide array of AKT inhibitors has been discovered through experimental and computational approaches. In this regard, herein we present a comprehensive overview of AKT inhibitors identified using computer-assisted drug design methodologies (including docking-based and pharmacophore-based virtual screening, machine learning, and quantitative structure–activity relationships) and successfully validated small molecules endowed with anticancer activity. Thus, this review provides valuable insights to support scientists focused on AKT inhibition for cancer treatment and suggests untapped directions for future computer-aided drug discovery efforts.

## 1. Introduction

AKTs are a group of serine/threonine kinases, also known as protein kinase B or PKBs, which play an important role in the regulation of a wide range of cellular functions, such as cell growth and proliferation, glucose metabolism, genome stability, transcription and protein synthesis, and neovascularization [1,2]. Specifically, this enzyme is a key component of the PI3K/AKTs/mTOR signaling pathway, whose overactivation contributes to the development of many human cancers and resistance to chemotherapeutic drugs [3,4].

Three different isoforms are composing the AKT family. The first isoform to be discovered and characterized was AKT1 (also known as PKBα) [5], followed by AKT2 and AKT3 (also named PKBβ and PKBγ, respectively) [6,7]. A differential tissue-specific expression and cellular localization is observed for the three isoforms. AKT1 is expressed ubiquitously in the cytosol, as well as at the plasma membrane, whereas AKT2 and AKT3 can be especially found in muscle tissue and in neurons, respectively [8]. Interestingly, the three isoforms play non-overlapping or opposing functions in pathological conditions. As an example, AKT1 has been shown to suppress breast cancer cell migration and invasion in in vitro studies, while AKT2 promotes these processes, potentially facilitating cancer metastasis [9,10]. These opposing roles of AKT1 and AKT2 in cell migration and invasion have also been validated in mouse models in vivo [11].

Structurally, the three isoforms share a generally conserved sequence (overall identity higher than 75%; Figure 1) and common structural organizations characterized by the presence of an N-terminal pleckstrin homology (PH) domain, an interdomain linker, a kinase catalytic domain, and a C-terminal hydrophobic motif. 

As expected, in the three isoforms the most conserved domain is the catalytic domain (90% of sequence identity), while significative lower conservation is observed in the interdomain linker (25% of sequence identity) [12]. AKT activation occurs following the conversion in the plasma membrane of phosphatidylinositol (4,5)-bisphosphate (PI2P) into phosphatidylinositol (3,4,5)-trisphosphate (PI3P) by phosphatidylinositol 3-kinase (PI3K) (Figure 2A). The PH domain recognizes the charged head of the PI3P thanks to the presence on its structure of a proper site called the PI3P-binding site, allowing the translocation of AKT from the cytosol to the plasma membrane and subsequently activating the phosphorylation of Thr308 and Ser473. These double phosphorylations lead to increased accessibility of the ATP-binding site in the catalytic domain, thus resulting in an improvement of the kinase activity [13,14].

The PH domain can also negatively regulate the function of AKT1. Indeed, the interaction of the PH domain with the catalytic domain generates an autoinhibited AKT form that prevents accessibility of the ATP-binding site to the ATP molecules.

This inactive closed conformation is generally called “PH-in” which is distinguished from the opened active conformation “PH-out” (Figure 2). Several types of AKT inhibitors have been developed as a result of the presence of three well-known ligand-binding pockets in the AKT1-3 structures: the ATP-binding site, the allosteric site, and the PI3P-binding site. Detailed structural information on these pockets has been reported in the literature for isoform AKT1 and is summarized herein. 

Regarding the ATP-binding site, no structures have been released of ATP bound to this pocket in the RCSB protein data bank (PDB) [15], but important clues can be gathered from the complex between AKT1 and adenylyl-imidodiphosphate (AMP-PNP; PDB ID: 4EKK [16]). Specifically, the ATP-binding pocket is composed of different structural elements that surround the central cavity, in which is accommodated the ATP substrate [16]. The hinge region (residues 227–230) connects the C-terminal and N-terminal domains and stabilizes the ATP adenine ring by two hydrogen bonds. Additionally, the glycine-rich loop (residues 157–162), the DFG motif (residues 292–294), and the αC-helix (residues 191–104) supply some amino acids which contribute to the constitution of the catalytic site. Compounds able to interact with this pocket prevent the binding of the ATP molecules, acting as competitive inhibitors. 

The PI3P-binding site is mainly composed of positively charged residues, in accordance with the specificity of the site for the PI3P head (i.e., inositol-1,3,4,5-tetraphosphate; 4IP). One of the crystal structures of the AKT1 PH-domain in complex with 4IP is available under the PDB ID 1UNQ [17]. The phosphate groups of 4IP established a complex network of hydrogen bonds and salt bridges with Lys14, Glu17, Tyr18, Ile19, Arg23, Arg25, Asn53, and Arg86. Also in this case, small molecules able to interfere with this site can prevent the binding event between AKT1 and PI3P, acting as competitive inhibitors and blocking the protein translocation. 

Finally, a third druggable site is formed in the surface involved in the protein-protein interaction between the kinase catalytic and PH domains. This pocket is only accessible in the “PH-in” conformation and can be targeted by AKT allosteric inhibitors. The binding of proper small molecules to this site allows for the (i) stabilization of the inactive conformation of AKT in which the accessibility of the ATP-binding site is reduced and (ii) stabilization of the PH-in conformation in which the PI3P cavity is shielded from the solvent and unavailable for PI3P recognition.

Following our interest in the kinase inhibitors’ field [18,19], we recently investigated the AKT1, reporting the identification of a novel ATP-site-directed AKT1 inhibitor with anticancer activity [20]. Even though, to date, no FDA-approved drugs against this protein are on the market, many efforts have been made in the discovery of molecules capable of targeting AKT by binding to one of its three pockets [4,13,21], with particular emphasis on small molecules for cancer treatment [3,4,14]. In this context, we herein provide a comprehensive literature survey of published papers reporting the use of in silico approaches to identify AKT-targeted small molecules with experimentally validated anticancer activity.

The collected works were classified based on the targeted site (i.e., ATP-binding site, allosteric site, or PI3P-binding site) and the main applied methodology, namely docking-based, pharmacophore-based, machine learning (ML), or quantitative structure-activity relationship (QSAR) as reported in Table 1. For instance, most of the examined manuscripts employed molecular docking experiments either alone or in combination. In particular, docking simulations were performed with three different aims: (a) to carry out virtual screening of compound libraries; (b) to refine the results obtained in pharmacophore-based, ML, or QSAR studies, and (c) to predict the binding mode of specific compounds. In light of this consideration, we classified the docking-based approach only to those works in which this computational technique was applied as the main methodology. Furthermore, three-dimensional (3D) pharmacophore models were also developed by applying two well-known strategies: (i) ligand-based pharmacophore modeling, which uses small ligands with different binding affinities to build predictive models in the absence of drug-target structural data, and (ii) structure-based pharmacophore modeling, which uses structure-based data and the bioactive conformations of known ligands to build models. Finally, for the best-characterized compound(s) reported in each work, we report the 2D structures and the available biological activities in terms of half-maximal inhibitory concentration (IC_50_), half-maximal effective concentration (EC_50_), half-maximal growth inhibition concentration (GI_50_), dissociation constant (Kd), inhibition constant (Ki) and percentage of inhibition (%inh). The reported activities can be referred to as assays performed on the isolated enzyme (i.e., AKT1, AKT2 or AKT3) or cellular system. In the latter case, the cellular line employed in the biological test is also indicated.

## 2. ATP-Binding Site

### 2.1. Docking-Based Approaches

From a historical point of view, the first published works were oriented towards targeting the AKT2 isoform.

In 2007, Donald et al. [22] started their work looking for a fragment that could act as a hinge-binder at the AKT2 ATP-binding site. Virtual screening of a fragment library composed of about 300,000 low molecular weight compounds (≤250 Da) allowed the selection of the 7-azaindole molecule **1** (Figure 3) as a possible hinge binder. Co-crystallization experiments performed with the PKA-AKT2 chimeras validated the docking predictions, and in its experimental binding mode, compound **1** established a double hydrogen bond interaction with the hinge residues (Glu121 and Ala123 in the PKA-AKT2 chimera, PDB ID: 2UVX [39]). Notably, the PKA-AKT2 chimera was previously validated as a valuable AKT2 surrogate for ATP-binder discovery [40]. Using **1** as the starting point, the authors coupled structure-based design and protein-ligand crystallography to rapidly identify novel potent AKT2 inhibitors (Figure 3). Specifically, to improve the synthetic accessibility of the scaffold and maintain the ability to form hydrogen bonds in the hinge region, the 7-azaindole core was replaced with a purine system. The fragment was suitably decorated and focused analogs were designed. Based on the crystallographic data provided by AKT2 bound to the known isoquinoline-5-sulfonamide inhibitor **2** (PDB ID: 2JDO [41]), the authors designed a series of derivatives (**3**–**5**, Figure 3) to mimic the interactions described as crucial between compound **2** and AKT2. The introduction of the basic amine in **3** enabled favorable polar contacts with the acidic residue Glu127 and the backbone carbonyl of Glu170. Furthermore, the addition of a terminal hydrophobic group (i.e., a benzyl moiety in compound **4**) resulted in a greater than 15-fold increase in AKT2 enzyme inhibition (IC_50_ 0.40 µM) compared to analog **3** (IC_50_ 6.9 µM). The final activity boost was obtained with the halogenation of the terminal aromatic ring (compound **5** with IC_50_ of 0.009 µM). The authors explained that the superior activity of **5** with respect to **2** was due to (i) the two hydrogen bonds interacting with the hinge region and (ii) the presence of a more rigid molecular structure bearing the same key pharmacophoric elements. Indeed, given the high number of rotatable bonds, the binding of compound **2** was associated with a higher entropic penalty. Unfortunately, despite the increased inhibitory potency, compound **5** showed low anticancer activity when tested in the PC3 cell lines. The authors hypothesized that the purine derivatives might have had a cellular permeability issue due to the low ratio of lipophilicity (CLogP) to topological polar surface area (TPSA). To overcome this limitation, they designed analog **6** which was predicted to have a more pronounced lipophilic character (ClogP: 4.81). The newly designed compound maintained a nanomolar IC_50_ and an optimal ligand efficiency (0.38 kcal mol^−1^ per non-H atom) against AKT2. Additionally, derivative **6** induced growth inhibition on different cancer cell lines in the low micromolar range (4.5, 5.7, and 8.7 µM in PC3, HCT116, and U87MG cell lines, respectively) and decreased the levels of pGSK3, pS6, and pFKHR proteins, that are downstream targets of the AKT2 pathway.

In 2009, Medina-Franco et al. [23] described the use of a consensus docking strategy in the structure-based VS, focusing on the selection of novel AKT2 inhibitors. Generally speaking, the consensus approach entails the use of more than one docking software/scoring function in the same VS pipeline to improve the overall docking performance [12,13]. Specifically, two docking software (i.e., FRED [42] and GOLD [43]) were used to explore AKT2 by using the two protein conformations co-crystallized with the known ATP-binding site inhibitors **AT7867** (PDB ID: 2UW9 [44]) and **A443654** (PDB ID: 2JDR [41]; Figure 4A). These structure-based strategies enabled the virtual screening of 105,937 lead-like compounds and the selection of nineteen putative AKT2 inhibitors that matched at least one of the following two criteria: top-ranked position based on the computed docking scores and/or the ability to make hydrogen bonds with the two key hinge residues Glu230 and Ala232. Notably, the co-crystallized inhibitors **AT7867** and **A443654** performed this double interaction (Figure 4A). Compound **7** (Figure 4B) was selected as the top-ranked molecule in the docking experiments against the 2UW9 protein model and was validated as a real hit in biological assays (IC_50_ = 1.1 µM). Interestingly, no interaction with the hinge residues was observed between this small molecule and the kinase domain. Conversely, the benzoxazolinone system performed two hydrogen bonds with the side chain atoms of Thr213 and Thr292, the 4*H*-1,4-benzoxazin-3-one moiety and Asp293 established a third hydrogen bond, and a π-π interaction emerged between the phenyl ring of **7** and Phe443. In the biological assays, compound **7** showed a pan-AKT inhibition activity with IC_50_ values of 2.6 µM, 1.1 µM, and 4.0 µM for AKT1, AKT2, and AKT3, respectively. Furthermore, this molecule exhibited interesting inhibition activities on the growth of MDA-MB-468 (EC_50_: 3.8 µM) and MDA-MB-453 (EC_50_: 10 µM) cancer cell lines. 

It is well known that competitive kinase inhibitors mimic the interaction of the ATP adenine group with the hinge region, generally through a heteroaromatic system that can ensure an effective binding to the ATP-binding site [45,46,47].

Starting from this knowledge, in 2015 Chuang et al. [24] filtered the SPECS commercial library to keep only compounds that (i) included in their structure heteroaromatic ring systems able to form hydrogen bonds with the protein and (ii) were commercially available. The retained subset (35,367 compounds) was docked against the AKT1 ATP-binding site (PDB ID: 3MVH [48]) using the DOCK program [49], and the results were filtered considering energy score values, visual inspection, and chemical diversity. Finally, forty-eight compounds were selected as virtual hits for biological validation. The in silico procedure led to the identification of compound **8** (Figure 5), which was able to inhibit about 75% of AKT1 activity at a concentration of 100 µM.

The compound was further characterized by cytotoxicity evaluation on HCT-116 human colon cancer cells and HEK-293 normal cells, in which the inhibitor showed IC_50_ values of 9.5 and 152.6 µM, respectively, resulting in a selectivity index (IC_50_-HEK-293/IC_50_-HCT-116) of 16.1. In the proposed binding mode, **8** established two hydrogen bonds with the AKT1 hinge residues, Thr211 and Ala230. This small molecule also showed multiple hydrophobic interactions with surrounding residues, including Leu156, Val164, Met227, Tyr229, Phe237, Met281, Phe438, and Phe442.

In 2022, three computer-aided drug discovery (CADD) approaches described the identification of new competitive AKT1 small molecule inhibitors endowed with anti-cancer activities [20,25,26].

In the first work, Zhong et al. [26] queried the ZINC collection to rationally identify novel AKT1 inhibitors from compounds of natural origin. The original database (ZINC15 natural collection) consisted of 17,931 commercially available molecules. LibDock module from Discovery Studio [50] was used to perform docking experiments (PDB ID: 4EKL [16]) and the twenty top-ranked hits were chosen for further in silico studies, e.g., absorption, distribution, metabolism, excretion (ADME), and toxicity predictions. Particular attention was paid to the aqueous solubility, the human intestinal absorption, and the cytochrome P450 2D6 inhibition. Additionally, the stability of the predicted ligand binding mode was assessed using molecular dynamics (MD) simulations. The two final candidates, **andropanoside** (**9**) and **neoandrographonide** (**10**) (Figure 5) were subjected to biological characterization by using the human osteosarcoma MG63 cell line and showed an effective ability in inhibiting the cell growth, although no explicit data values (e.g., IC_50_ or EC_50_) were provided. Additionally, both compounds inhibited the AKT1 expression in MG63 cells at a concentration in the submicromolar range. However, the authors did not provide direct evidence of AKT1 inhibition and/or binding for the two molecules.

In the second work performed by Noser et. al., docking studies were applied for the identification of a PI3K/AKT1 dual inhibitor [25]. The advantage of developing dual inhibitors of AKT1 and PI3K is that both proteins are involved in the same signaling pathway and play critical roles in the proliferation and survival of cancer cells. Therefore, targeting these two proteins simultaneously can lead to a more potent and efficient anticancer effect compared to targeting either protein alone. The authors used the PyRx platform [51] to virtually screen a small set of twenty-one in-house compounds for the ability to interact with both AKT1 and PI3K. The results highlighted that compound **11** (Figure 5) had favorable binding energy against both PI3K and AKT1 targets and promising in silico drug-like properties. In the predicted binding mode, the compound interacted with the AKT1 hinge region by hydrogen bonding Ala230, Glu228, and Thr195, while a π-cation interaction was predicted with the basic side chain of Lys179. Unfortunately, no information about the ability of **11** in inhibiting the isolated enzymes was provided, but the molecule was able to reduce the phosphorylation levels of both AKT1 and PI3K in a dose-dependent manner in the Caco colon cancer cell line. This small molecule also inhibited tumor growth in A549, MDA-231, Caco, PCL, and MCF-7 cancer cell lines with IC_50_ values of 40.91, 38.45, 23.34, 56.33, and 50.15 μM, respectively, and was characterized by low cytotoxic effects on WISH normal cells (IC_50_ = 124.4 μM). 

Finally, we have recently reported the identification of novel competitive AKT1 inhibitors as possible agents against acute myeloid leukemia (AML) [20]. The serendipitous discovery of compound **12** (Figure 6) as an AKT1 inhibitor prompted us to carry out modeling studies to aid the selection of **12**-like compounds for biological testing. As the first step, **12** was used as a query molecule in the BioSolveIT Feature Trees (FTrees) [52] software, a fast 2D-similarity screening tool used to mine a library of in-house compounds. The resulting compounds were then docked with Glide [53,54] using the AKT1 conformation co-crystallized with the clinical trial inhibitor **capivasertib** (PDB ID: 4GV1 [55]). This protocol led to the identification of the 5,6,7,8-tetrahydrobenzo[4,5]thieno[2,3-*d*]pyrimidin-4(3*H*)-one derivative **13** (Figure 6, **T126** in the original paper) as the most interesting compound. Based on the predicted binding pose generated by the docking protocol and subsequent MD simulations, **13** interacted with the hinge region by using the para-hydroxyl group of the catechol moiety, while the second hydroxyl group was in contact with Thr291. Interestingly, the endocyclic amide established well-conserved, water-mediated interactions with residues Thr291 and Asp292. The compound inhibited the AKT1 activity with IC_50_ and K_i_ values of 1.99 and 0.41 μM, respectively. Accordingly, a clear effect on the growth inhibition and induction of apoptosis in AML cells at low micromolar concentrations was observed. Indeed, **13** showed IC_50_ values 4.2, 4.3, 2.4, 9.2, and 6.9 µM for OCI-AML3, IMS-M2, OCI-AML2, MOLM-13, and SKM-1 cell lines, respectively. Additionally, given the presence of a catechol moiety that is counted as one of the pan-assay interference compounds’ (PAINS) motifs, we performed specific experiments to rule out the possibility that the compound interfered with the biochemical and cellular assays by the mechanism of intrinsic fluorescence effect, metal chelation, and chemical aggregation. Because of these encouraging data, the biologically validated hit **13** is now the subject of additional biological investigations and hit-to-lead efforts.

Still, in 2022 Zhu et al. [27] reported a new potent and selective AKT1 proteolysis targeting chimera (PROTAC). PROTAC is a heterobifunctional molecule made up of two heads and a linker that can be used to degrade undesirable proteins. Indeed, while one PROTAC head binds to the target protein intended for degradation, the second head interacts with the E3 ubiquitin ligase, leading to ubiquitination and proteasome-mediated degradation of the former protein [56]. In this context, structure-based approaches are especially useful in the rational design of the PROTAC linker, where they may support the choice of the preferred chain length and/or composition. In particular, computational protocols, such as the “protein-protein docking & double clustering” method [57] can be applied in the prediction of PROTAC-mediated ternary complex model (i.e., PROTAC + E3 ligase + target protein). Using this strategy, the authors aimed to create a PROTAC molecule by linking the known AKT1 inhibitor **14** and the E3 ubiquitin ligase binder **pomalidomide** (Figure 7). Among the designed and tested compounds, derivative **15** (Figure 7) rapidly and completely removed AKT1 protein, also exhibiting efficacious antiproliferative effects on haematological cancer cells. From a chemistry point of view, this PROTAC molecule was characterized by a phenyl ring in the linker that was able to establish a favorable π-π interaction with the Phe236 of AKT1. The authors suggested that **15** and its analogues could be valuable for studying AKT1 biological functions and developing drugs to treat AKT-associated human cancers. Further evaluation of **15**‘s safety benefits and therapeutic potential in mantle cell lymphoma (MCL) treatment is ongoing.

### 2.2. Pharmacophore-Based Approaches

In 2011, Dong et al. [28] reported the first successful example of AKT1-targeting molecules for cancer that was identified by ligand-based pharmacophore modeling. The authors collected, from the literature, seventy-four competitive AKT1 inhibitors with IC_50s_ spanning over six orders of magnitude (from 0.00016 to 32.25 µM). This library was then split into twenty-four and fifty compounds constituting the training- and the test set, respectively. The former set was used to generate a HypoGen [58] model composed of four pharmacophoric features, including one hydrophobic group, one hydrogen bond acceptor, and two hydrogen bond donors (Figure 8; compound **16** is reported as a representative of the training set). The model showed an area under the ROC curve (AUC) of 0.957, and a correlation coefficient of 0.803 between the experimental and predicted AKT1 IC_50s_ of the compounds in the test set. Encouraged by these results, the authors used the developed pharmacophore to screen 1024 and about 60,000 compounds from an *in-house* and the Maybridge databases, respectively. Among them, eighty candidates showed a good fit with the pharmacophoric features, and their selection was refined by using molecular docking experiments. To select both the proper docking algorithm and protein conformation, the authors performed self-docking studies on six different AKT1 protein structures (i.e., PDB IDs: 3CQU [59], 3CQW [59], 3MV5 [48], 3MVH [48], 3OCB [60], and 3OW4 [61]) employing five molecular docking software (i.e., Libdock [50], LigandFit [62], FlexX [63], CDOCKER [64], and flexiDOCK [65]). The best result was obtained by combining the 3OCB conformation with the flexiDOCK software (average RMSD in self-docking studies of 0.11 Å). Nine compounds were finally selected after docking studies and subsequently subjected to biochemical assays (HTScan PKB/Akt1 Kinase Assay Kit) and cytotoxicity studies on PC3, OVCAR-8, and HL-60 cell lines. The best inhibitory activities against AKT1 were shown by the flavonoid compound **17** with an IC_50_ value on the isolated AKT1 of 5.4 μM (Figure 8). The cellular assays also disclosed promising antiproliferative activity concerning all tested cell lines (PC3M, OVCAR-8 human ovarian carcinoma cells, and HL60 human leukemia cell line), with an IC_50_ ranging from 2.5 to 23.8 μM. Additionally, the apoptotic ability of **17** toward the OVCAR-8 cell line was tested as well. After 72 h of exposure, induction of apoptosis was observed in 40.19% of the treated cells. As proposed by the authors, computational studies suggested two main interactions between the flavonoid compound and AKT1: (i) the catechol moiety formed two hydrogen bonds with the residues Ala230 and Glu228, and (ii) the 5-hydroxyl group of the carboxylic oxygen of the chromanone ring interacted with the Lys179 residue. In an effort to enhance the inhibition activity of this chemical family, **17** was submitted to chemical optimization [31]. Based on docking and MD simulations, only derivative **18** was selected for further studies, as this compound was predicted to establish an additional hydrogen bond involving the 7-hydroxyl group and a π-π stacking interaction between one phenyl functionality and the Phe161 residue (Figure 8). Unfortunately, derivative **18** did not show any improvement compared to the parent compound **17** when submitted for biological experiments.

In a sequent study published in 2013 [30], the same research group applied the in silico workflow developed to identify compound **17** to investigate analogues of the previous published compound **5** (Figure 3). Specifically, a series of diphenyl methylamine derivatives were rationally designed in a stepwise manner by first decorating the primary amine with substituents of a different nature, and then replacing the purine system with a pyrazole ring. The generated molecules were submitted to pharmacophore screening and molecular docking studies. The best analog produced by this work was compound **19** (Figure 8), endowed with an IC_50_ of 0.038 μM on isolated AKT1 and low micromolar anti-proliferative activity on OVCAR-8, HL60, and HCT-116 cancer cell lines (8.1, 5.3, and 8.9 μM, respectively). Additionally, kinase selectivity studies underlined the excellent selectivity of this derivative against Aurora A, Drak, IKKb, GSK3b, SYK and JAK2 kinases. Finally, the information gathered by the produced biological results was used to refine the original pharmacophore model, shown in Figure 8 for the AKT1 inhibitor.

Unlike the previous ligand-based approach, in 2020 Fratev et al. [29] proposed a structure-based pharmacophore model which was developed by combining the application of the e-Pharmacophore tool [66] from Schrödinger (https://www.schrodinger.com/), with fragment-based virtual screening. Specifically, the 3D model was generated based on docking results obtained for several hundred fragments oriented into the ATP-binding site of AKT1 (PDB ID: 3QKK [67]). The position of the well-scored fragments was used to define a specific pharmacophore feature (i.e., hydrogen bond acceptor, hydrogen bond donor, hydrophobic, negative ionizable, positive ionizable, or aromatic ring) according to the chemical nature of the fragment. The best-obtained model included four aromatic rings, three hydrogen-bond acceptors, and two hydrogen bond donors, and was integrated into a virtual screening protocol to query 3.5 million “lead-like” ZINC compounds. The top 1% of retrieved compounds (i.e., 35,000 ligands) were then docked and rescored by using Glide-SP. Finally, nine candidates emerged as virtual hits and were subjected to biochemical and cellular assays. Despite the efforts, the most interesting compound (**20**; Figure 9) was endowed with only a low ability to inhibit AKT1 activity and cancer cell growth. No information was provided about the fitting of derivative **20** on the developed pharmacophoric model.

### 2.3. Machine Learning Approaches

In 2021, Wang et al. applied a deep learning strategy called SyntaLinker to design novel AKT1 inhibitors [32]. SyntaLinker connects molecular fragments using syntactic pattern recognition and deep conditional transformer neural networks to design compound libraries by learning from the vast knowledge contained in chemogenomic repositories, such as ChEMBL [68]. This process enables a “scaffold hopping” approach for the identification of novel molecules with a potentially high binding affinity against the selected target. The authors started by decomposing the structure of the known AKT1 inhibitor **capivasertib** into three parts: a central piperidine scaffold, the pyrrolopyrimidine ring, and the chlorophenyl substituted propanol as terminal fragments (TFs) (Figure 10). They then applied the SyntaLinker strategy and generated new scaffolds able to link the two TFs as well as to satisfy some structural filters (e.g., maximal bond distance ranging from three to seven atoms and only one aromatic ring). The retrieved compounds were submitted to docking experiments leading to the selection of twenty-four molecules with ligand binding that have similar poses to **capivasertib**. Among these molecules, the authors selected the aminobenzamide derivative **21** (Figure 10) for synthesis and kinase activity validation, obtaining a moderate IC_50_ value of 7.2 μM. Subsequent chemical optimization strategies were carried out by (i) replacing the hydroxyethyl group with a basic methylamine moiety and (ii) removing the amine between the benzene ring and the pyrrolopyrimidine to shorten the length of the molecule. These modifications (compound **22**; Figure 11) significantly improved the inhibitory activities of AKT1, AKT2, and AKT3 (i.e., IC_50_s of 0.088, 0.7, and 0.092 μM, respectively). The predicted pose of **22** in the ATP binding site of AKT1 showed that this inhibitor performed the conserved hydrogen bond interactions with the hinge residues Glu228 and Ala230, and an additional hydrogen bond with Asn279. Furthermore, hydrophobic interactions with Val164, Phe161, and Leu181, were present. Of note, **22** exhibited selective anticancer activity against U937 (IC_50_ value of 0.39 μM) cancer cells, while displaying less potent activities against other cancer cell lines (activities against HEPG-2, HEK293 and HCT116 with IC_50_ values of >10 μM, 1.96 μM, and 2.25 μM, respectively).

### 2.4. QSAR Modelling

In 2015, Zhan et al. [33] described the development of a molecular docking-based QSAR model for AKT1 inhibitors. The model was trained with forty-seven molecules collected from the literature data by applying the support vector regression (SVR) method. The collected set was subjected to docking simulations using the LigandFit program [62] and several scoring functions were employed to calculate the ligand binding scores. Additionally, key interaction profiles referred to the distances between the docked compounds and specific amino acid residues in the AKT1 were derived for each compound pose. Examples of these key amino acid residues included Glu228, Ala230, Glu234, Glu292, and Phe163. The combination of docking scores, key interaction profiles, and molecular descriptors (e.g., clogP and tPSA) allowed the generation of a QSAR model for bioactivity estimation with a very interesting predictability (R^2^ = 0.934) that encompassed the predictability of the single LigandFit scoring function (R^2^ < 0.417). The generated model was then used to select the most promising AKT1 inhibitors within a virtual library of 4-aminopyrimidine derivatives. Compound **25** (Figure 11) emerged as the most promising virtual hit (predicted pIC_50_: 7.9), and its experimental activity on the isolated enzyme (IC_50_: 7.7 nM; pIC_50_: 8.1) confirmed the in silico prediction. Additionally, this small molecule showed promising antiproliferative effects against the HCT116 and OVCAR-8 cancer cell lines (IC_50_ of 5.15 and 22.67 μM, respectively). According to the docking studies reported by the authors, the molecular interactions between **25** and AKT1 involved two hydrogen bonds between the compound 4-aminopyrimidine group and Glu228 and Ala230 residues. Moreover, a hydrophobic interaction was hypothesized between the 4-chlorophenyl ring and the amino acid Phe161. Finally, an ionic interaction was predicted between the primary amino group of **25** and the Glu292 residue.

## 3. Allosteric Site

### 3.1. Docking-Based Approaches

Two papers published in 2022 described the application of docking-based virtual screening to discover potential allosteric AKT1 inhibitors. The SiBioLead web-tool (https://sibiolead.com/) was used in both works to identify a single lead molecule able to act as a dual inhibitor.

The first study, published by Al Shahrani et al. [35], was aimed at finding new molecules that could overcome resistance to **vemurafenib** in the treatment of melanoma. **Vemurafenib** is an approved kinase inhibitor that specifically inhibits the activity of the mutated form of the BRAF protein, known as BRAF-V600E [69]. As resistance to this drug is often due to reactivation of the MAPK and PI3K/AKT signaling pathways, the researchers planned to identify a dual inhibitor that could target both BRAF-V600E and AKT1 proteins. KINAcore and KINA-set libraries from ChemBridge were selected as the compound source and merged to create a collection of 23,365 compounds sharing 3D pharmacophore fingerprints with well-validated and published kinase inhibitors. The virtual screening of the two libraries against the BRAF-V600E (PDB ID: 1UWH [70]) led to the selection of the top 15 compounds for which more stringent docking calculations were produced, among which the evaluation of the potential affinity for the AKT1 target (PDB ID: 6HHG [71]). Compound **244** (Figure 12) was found to potentially bind both BRAF-V600E and AKT1 with high binding energy and affinity.

Specifically, docking experiments predicted a conserved π–π stacking interaction with Trp80, and three hydrogen bonds with Ser205, Asn54 and Gln79. Biological validation showed that this small molecule dose-dependently inhibited BRAF-V600E and AKT1 with IC_50_ values of 635 and 154.3 nM, respectively, and also controlled the proliferation of normal (A375-N cell line) and **vemurafenib**-resistant (A375-R cell line) melanoma cells. Indeed, while the effect of **vemurafenib** was reduced by about 3-fold in resistant cells (GI_50_ of 13.73 and 34.60 µM for A375-N and A375-R, respectively), no activity decrease was observed in the case of treatment with **24** (GI_50_ of 222.3 and 230.5 nM for A375-N and A375-R, respectively)**.** Additionally, the compound was able to induce cell cycle arrest and apoptosis and to reduce the number of cells with high levels of pERK and pAKT, a condition generally associated with insurgency of resistance mechanisms.

In the second study, Abohassan et al. [34] reported their efforts in identifying a single lead molecule able to inhibit both PI3K and AKT1 for the treatment of AML. Because the majority of potent kinase inhibitors have molecular weights between 350 and 750 Da, the authors built a chemical library for virtual screening composed of about 800,000 ChemBridge compounds respecting this range. To accelerate the screening process, diversity-based high-throughput virtual screening (D-HTVS) was applied. In the first step, structurally diverse compounds selected from the original library were docked against the target protein, with each compound taken as a representative of a given scaffold. In the second step, each scaffold of the top 10 compounds was used to collect a focused library for further docking simulations. First, the D-HTVS workflow was applied to explore the PI3Kγ protein. Indeed, PI3K is expressed in several isoforms, and previous studies indicated that the molecules targeting both PI3Kγ and PI3Kδ subunits showed maximum efficacy in various cancers where PI3K is implicated [72]. Then, the obtained virtual hits were subjected to a funnel-like process in which docking experiments were used to evaluate the affinity of the queried library again PI3Kα, PI3Kβ, PI3Kδ, and AKT1, each time discarding the molecules that did not meet the required criteria. Specifically, the authors selected only compounds with high affinity against PI3Kγ, PI3Kδ, and AKT1 and low affinity against PI3Kα and PI3Kβ. Four molecules matched these criteria, with compound **27** (Figure 12) chosen as the most promising hit based on the binding energies and docking poses calculated against all the protein targets considered in this study. Regarding the predicted interactions with AKT1, the benzo[*de*]isoquinoine system of **25** formed a π–π stacking interaction with Trp80 as well as additional hydrophobic contacts with Leu264, Ile84, Val270, and Arg270 residues. No intermolecular hydrogen bonds were predicted, while one intramolecular hydrogen bond was hypothesized between the nitrogen atom of the aminic linker and one of the oxygens of the barbituric ring. The in vitro kinase assays corroborated with computational predictions, showing that **25** inhibited PI3Kγ, PI3Kδ, and AKT1 kinases in the nanomolar range. Of note, this small molecule also showed potent anticancer activity when tested against THP-1 and HL-60 AML cell lines, validating the hypothesis that dual inhibitors targeting PI3K and AKT can efficiently inhibit AML cell proliferation.

### 3.2. Pharmacophore-Based Approaches

In 2017, Lakshmi et al. [36] gathered thirty-six known AKT1 pyridopyrimidine biphenyl derivative inhibitors (exemplified by compound **26**, Figure 13) to create a 3D pharmacophore model with Phase [73] composed of three hydrogen-bond acceptors, two positive groups and two aromatic rings. This model was then used to screen a library of about 5000 natural compounds and the 708 retained molecules were docked into the allosteric site of AKT1 (PDB ID: 3O96 [74]). Specifically, the authors employed a funnel-like pipeline, including Glide [54] HTVS, SP, XP, and Prime [75] MMGBSA steps. This process resulted in the retrieval of forty-five compounds, which were visually inspected leading to the final selection of twenty-three molecules able to potentially interact with the residues Ser205 and Trp80. Further MD computational analysis, based on the ligand hydrogen bond occupancy with key kinase residues, persistent polar contacts, and favorable binding free energy, suggested quercetin-7-O-*d*-glucopyranoside (**27**, Figure 13) as the most promising virtual hit. Biological validation of **27** revealed that this small molecule induced dose-dependent inhibition of breast cancer cells (MDA MB-231) and down-regulated the expression of p-AKT1 (Ser473). The authors further confirmed the ability of **29** to bind protein kinase AKT1 by measuring a Kd of 0.246 μM.

## 4. PI3P-Binding Site

### Pharmacophore-Based Approaches

In 2008, Mahadevan et al. described an in silico strategy for the development of PI3P site binders starting from the crystal structure of the AKT1 PH domain (PDB ID: 1H10 [76]) in complex with the head of PI3P (i.e., 4IP) [37]. The intermolecular interactions performed by the ligand with the protein allowed the generation of a pharmacophore query generated with UNITY (Tripos, L.P.) [77], although the authors did not explicitly describe the selected pharmacophore features. The generated pharmacophore query was used to screen the National Cancer Institute (NCI) database, and the retrieved compounds were submitted to docking simulations. Based on the predicted binding mode, molecule **28** (Figure 14) emerged as the most promising AKT1 potential binder as it was able to mimic the natural substrate. Indeed, the ligand sulfonamido and the diazopyrazolyl moieties formed hydrogen bond interactions with Arg86 and Arg23, respectively, which are two residues already known to establish strong contacts with the phosphate groups of PI3P. Additionally, other hydrogen bonds were hypothesized between the backbone of Ile19 and Asn53 with the sulfonamide function of molecule **31**. By using surface plasmon resonance (SPR) experiments, the authors validated the molecule as a true binder of the AKT1 PH domain with a K_d_ value of 0.37 µM. Additionally, this PH-domain binder inhibited the interaction of PI3P to the PH domain of AKT1 with an IC_50_ of 0.08 µM. In cellular assays, compound **31** inhibited both the AKT1 phosphorylation with IC_50_ values of 4 and 13 µM in NIH3T3 and HT-29 cell lines, respectively, and the growth of HT-29 cells with an IC_50_ of 24 µM. Unfortunately, no appreciable effects were observed in in vivo experiments, probably due to the rapid metabolism/elimination of this compound.

In 2009, the same research group (Moses et al. in 2009) [38] applied the pharmacophore-based approach described above to query a library of about 300,000 compounds derived from different compound collections (NCI Chemical and Natural Products Library, the Maybridge Available Chemicals Directory, and the Lead-Quest Chemical Library). The best 20 compounds of each collection were kept, clustered based on the 2D structure, and reduced to 4 virtual hits submitted for biological evaluation. All the selected compounds showed a binding affinity for the PH domain of AKT1 in the low micromolar range, with K_d_ values ranging from 0.39 to 6.27 µM. Even though the four compounds belonged to different chemical families, they shared the presence of at least one sulphone containing a linker that connected two aromatic rings. With the aim to expand the structure-activity relationship of each chemical family, commercially available analogues of each one of the four validated hits were selected for further testing, but only the N-(1,3,4-thiadiazol-2-yl) benzenesulfonamide derivatives emerged as promising scaffolds in the development of PI3P-site binders. In particular, compound **29** (Figure 14) showed a K_d_ value of 0.45 µM, but no evident effects of inhibiting the recognition of PI3P (K_i_ > 50.0 µM) and preventing tumoral cell survival were observed (Table 2). The authors hypothesized that the inactivity of **29** in cellular assays could be imputed to the low predicted LogP and Caco-2 permeability of the compound, and thus they rationally designed some analogues to overcome this issue.

Docking studies of the hit binder performed with GOLD [43] suggested a complex network of polar interactions established with Lys14, Glu17, Arg23, and Arg25 (Figure 15). Interestingly, the phenyl moiety protruded the para-amino substituent towards the solvent and this position was explored by introducing modifications aimed to increase the calculated LogP and Caco-2 permeability. The best results were obtained for derivative **30** (Table 2) by adding a hydrophobic moiety, obtaining an improved K_i_ and cellular activity. To rule out the hypothesis that analogue **30** could be embedded into the cellular membrane and, subsequently to the cleavage of the amide bond, released into the cytoplasm, the non-cleavable compound **31** (Table 2) was synthesized.

Despite the predicted high LogP and low Caco-2 permeability properties, this derivative exhibited the most effective reduction of cell phospho-Ser473-AKT and cellular growth. Additionally, by using a fluorescent analog of **30**, the authors demonstrated that in the cellular environment, the compound was mainly located in the cytosol and/or lipid vesicles, potentially trapping AKT in the former matrix. Finally, this PI3P-site binder was also tested for antitumor activity against BxPC-3 pancreatic cancer xenografts at a dose of 125 mg/kg i.p., twice a day for 5 days, and demonstrated significant antitumor activity, with complete cessation of tumor growth and even regression during treatment.

## 5. Conclusions

In the era of precision oncology, AKT represents an attractive therapeutic target for the discovery of pathway-based targeted therapies selectively hitting cancer cells characterized by aberrant activation of PI3K/AKTs/mTOR signaling pathway.

Against this backdrop, we have conducted a literature survey focused on computer-aided approaches applied to the rational identification of AKT-targeting small molecules endowed with anticancer activity. Several considerations can be drawn from the analysis of the collected works and are listed below.

First, molecular docking and pharmacophore modeling emerged as the most commonly employed methods, whereas other approaches such as the development of QSAR models were less prevalent. Additionally, although ML algorithms, such as support vector machines (SVM) and deep learning models, have shown promise in predicting the activity and the properties of potential inhibitors [78,79], in the context of AKT for cancer, these approaches have been relatively underexplored. In this regard, accessing and analyzing the vast amount of data available in databases, such as PubChem and ChEMBL, becomes crucial. For instance, our research group recently released a KNIME workflow that can effectively aggregate information on the biological activities of compounds from different public/proprietary depositories. As a case study, we reported the application of such a tool for the generation of a curated dataset of over 358,000 compounds associated with experimental activity against AKT1 [80]. Access to such a wealth of data can support the construction of ML models and, more broadly, the use of ligand-based approaches to aid the development of novel AKT1 inhibitors.

Second, most of the reviewed studies focused on hit identification rather than hit-to-lead optimization. Specifically, the newly described hit compounds showed IC_50_ values ranging from 0.54 to 7.2 µM. Although these activities fall significantly short of the low nanomolar range exhibited by AKT1 inhibitors in clinical trials (Table 3; Figure 10 and Figure 15), it is important to acknowledge that the latter are the results of extensive optimization efforts. Nevertheless, computer-aided drug design methods can strongly support lead identification as shown by Wang et al. [32] in the successful optimization of compound **22** (IC_50_: 7.2 µM) to obtain derivative **23** (IC_50_: 88 nM).

Third, we assessed whether the described studies had evaluated, in silico and/or in vitro, the propensity of the newly identified hit compounds to form aggregates or exhibit promiscuous activity as PAINS. In this context, in the last years, several computational tools have been developed to mitigate the selection of false positives resulting from the interference of the small molecule with the biological assay rather than the specific ligand interaction with the target protein. For instance, the aggregator advisor [81] and SwissADME [82] servers represent two valuable freely accessible web interfaces that allow the recognition of potential aggregators or PAINS, respectively. Regrettably, the majority of the analyzed studies failed to address this critical aspect. Therefore, we attempted to scrutinize the compounds featured in this review with the two previously mentioned servers, noting that several molecules were flagged as potential aggregators (cLogP ≥ 5 and Tanimoto similarity with known aggregators ≥ 0.5: compounds **11** and **24**) or PAINS (presence in the chemical structure of (i) azo group: compounds **11** and **28**; (ii) catechol moiety: compounds **13**, **17**, **18**, and **27**; (iii) barbiturate ring: compound **25**). Against this backdrop, we would like to remind researchers that it should be standard practice to first carefully predict these compound’s characteristics and then conduct specific orthogonal tests to rule out artifacts that may arise during biological evaluations.

Fourth, of particular interest, is the absence of publications reporting on the computer-assisted design of covalent inhibitors. It is worth noting that two specific cysteine residues, well conserved among the three AKT isoforms (corresponding to Cys296 and Cys310 in AKT1), are located in the proximity of the allosteric site and have been already exploited in the identification of covalent inhibitors [8,71,83,84,85]. These small molecules possess both the selectivity resulting from targeting the allosteric pocket and the prolonged target residence time ensured by the covalent modification of the two non-catalytic cysteines in the AKT activation loop. As a result, this class of inhibitors is more effective at inhibiting AKT activity than non-covalent binders [86,87].

Fifth, it has been demonstrated that inhibiting both AKT1 and AKT2 is essential for obtaining a greater sensitization of tumor cells to apoptotic stimuli and reduceing AKT phosphorylation in vivo [88,89]. Despite this evidence, most of the papers reviewed here have tested their molecules only on one of the two isoforms, thus not considering the validated aspect of the dual AKT1/2 inhibition.

Sixth, our examination reveals that there are considerably more publications focused on the rational identification of competitive inhibitors as compared to allosteric inhibitors. While the abundance of research on competitive inhibitors is not surprising, given their historical prominence, it is important to recognize the potential advantages offered by allosteric modulation.

Finally, protein degraders have emerged as a promising class of therapeutics in drug discovery [56,90,91,92]. Even though some attempts have been made to discover AKT-targeted degraders [93,94,95,96], there is still ample room for further research and development that can be accelerated by the support of CADD approaches.

In conclusion, while several computational strategies have supported multidisciplinary efforts aimed at the rational identification of AKT inhibitors as anticancer agents, there remain intriguing avenues for future in silico research, particularly in the design of covalent inhibitors or small molecule degraders. These strategies offer promising possibilities for overcoming drug resistance and expanding the scope of personalized medicine in cancer treatment. In support of these objectives, the RCSB PDB is a valuable source of 3D data for structure-based CADD investigations. To provide additional support to the scientific community interested in the AKT target, we report here a comprehensive table (Table 4) summarizing the available crystal structures of AKT, either unbound or bound to a ligand, organized according to the isoform and the binding pocket.

## Figures and Tables

**Figure 1 pharmaceuticals-16-00993-f001:**
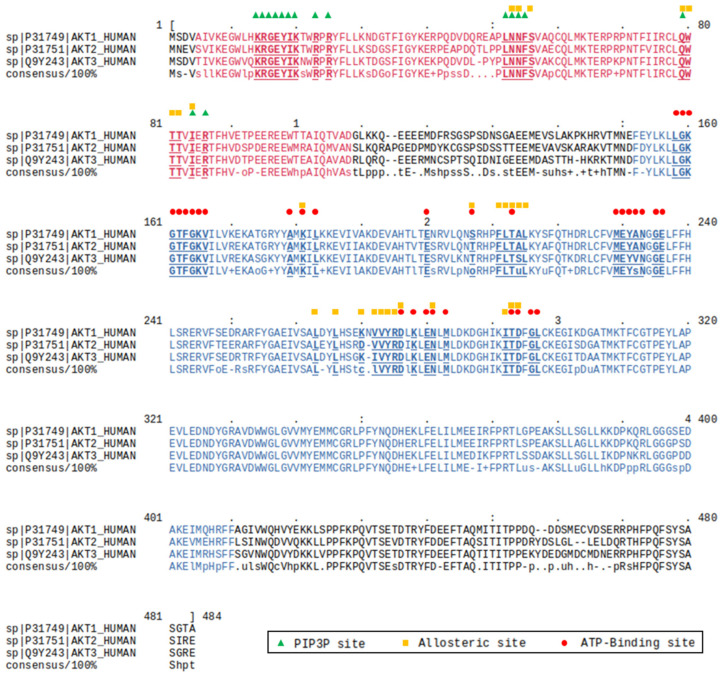
Sequence alignment of the three isoforms of AKT, i.e., AKT1, AKT2 and AKT3. The residues composing a ligand binding pocket are represented in bold and underlined. The specific pockets are differentiated by using different symbols. The PH domain and the kinase catalytic domain are color-coded in purple and blue, respectively. In the consensus sequence, conserved residues are indicated as capital letters, while non-conserved residues are represented as follows: c, charged; h, hydrophobic; l, aliphatic; o, alcohol; p, polar; s, small; t: turnlike; u, tiny; +, positive; -, negative. The alignment was performed by using the Cluster Omega server (https://www.ebi.ac.uk/Tools/msa/clustalo/ accessed on 1 June 2023).

**Figure 2 pharmaceuticals-16-00993-f002:**
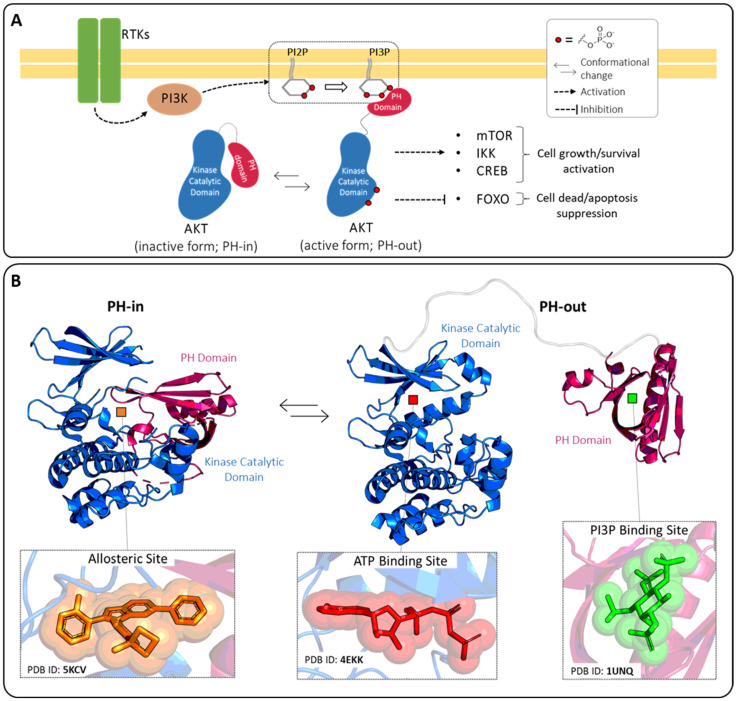
(**A**) Mechanisms of AKT regulation. The stimulation of receptor tyrosine kinases (RTKs) leads to activation of phosphatidylinositol 3-kinase (PI3K) and the subsequent conversion of the phosphatidylinositol (4,5)-bisphosphate (PI2P) into the phosphatidylinositol (3,4,5)-trisphosphate (PI3P). The inactive form of AKT (PH-in) is then recruited from the cytosol and translocated to the membrane thanks to the interaction between the AKT PH domain and PI3P. At this point, the double phosphorylation of AKT fully activates the protein (PH-out) that in turn modulates the downstream signaling proteins such as mammalian target of rapamycin (mTOR), IkappaB kinase (IKK), cAMP response element-binding protein (CREB) and Forkhead box O (FoxO). (**B**) AKT closed “PH-in” and opened “PH-out” conformations, with highlighted the three known ligand binding sites.

**Figure 3 pharmaceuticals-16-00993-f003:**
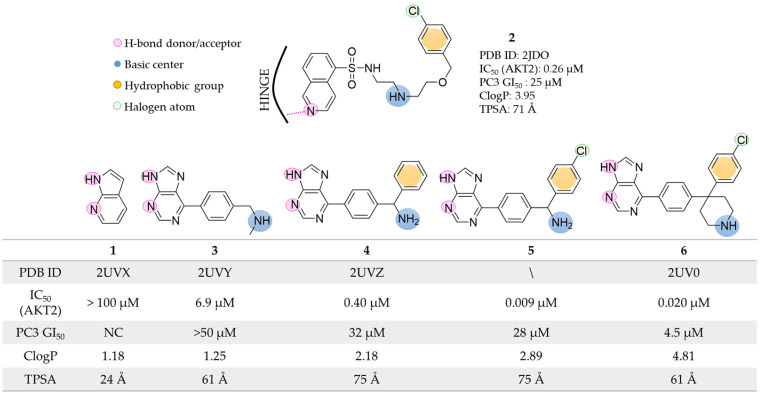
Targeting ATP-binding site. Rational identification of derivative **6** starting from fragment **1**. The AKT2-ligand interactions emerged from co-crystallization experiments are highlighted.

**Figure 4 pharmaceuticals-16-00993-f004:**
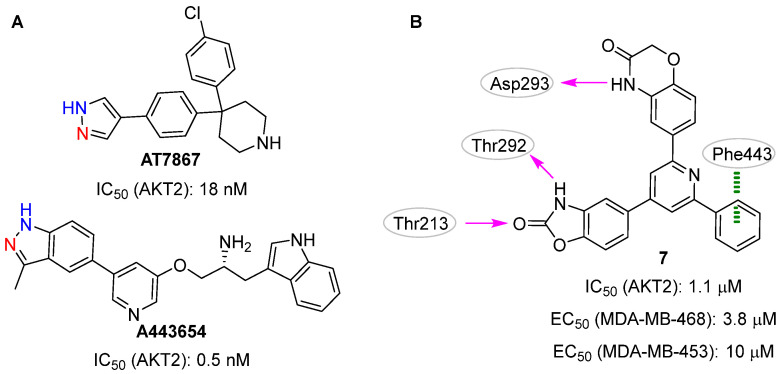
Targeting ATP-binding site. Co-crystallized AKT2 inhibitors **AT7867** and **A443654** (**A**) and validated hit **7** (**B**). The atoms involved in the hydrogen bond interactions with the AKT2 hinge residues Glu230 and Ala232 are highlighted in blue and red, respectively. Predicted polar intermolecular interactions are illustrated as follows: magenta arrow, hydrogen bond; green dotted line, π-π interaction.

**Figure 5 pharmaceuticals-16-00993-f005:**
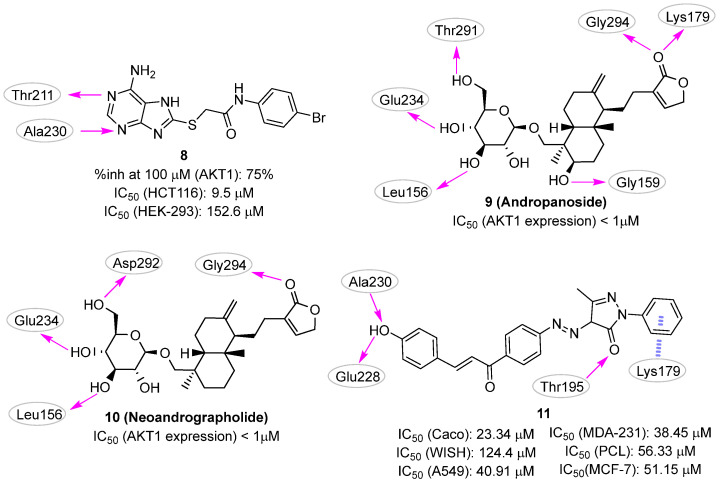
Targeting ATP-binding site. Validated hits **8–11** with the predicted polar intermolecular interactions illustrated as follows: magenta arrow, hydrogen bond; purple dotted lines: π-cation interaction.

**Figure 6 pharmaceuticals-16-00993-f006:**
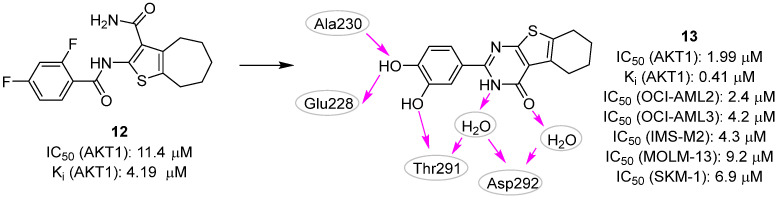
Targeting ATP-binding site. Validated AKT1 inhibitors **12** and **13** with the predicted polar intermolecular interactions for the latter compound illustrated as follows: magenta arrow, hydrogen bond.

**Figure 7 pharmaceuticals-16-00993-f007:**
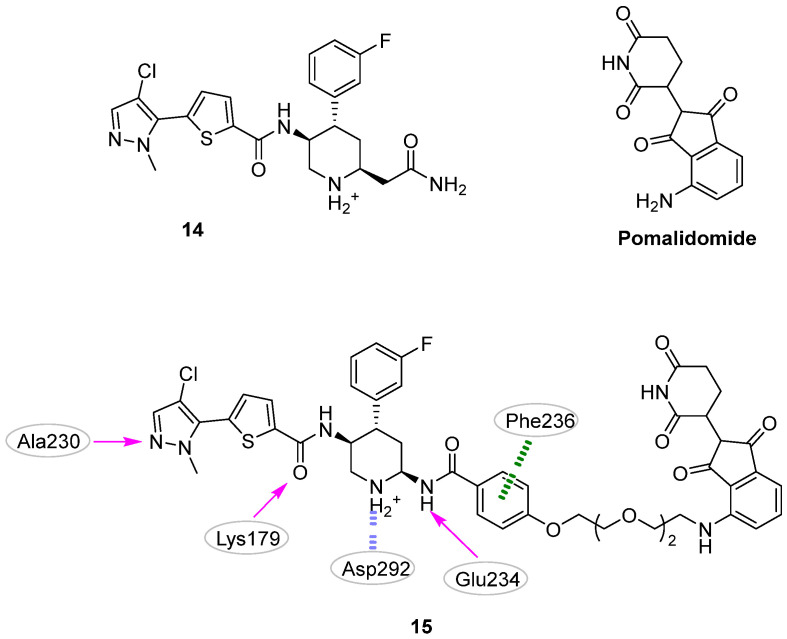
Targeting ATP-binding site. Compound **14** and **pomalidomide** were linked to generate the PROTAC molecule **15**. The predicted polar intermolecular interactions are illustrated as follows: magenta arrow, hydrogen bond; green dotted line, π-π interaction; purple dotted lines: π-cation interaction.

**Figure 8 pharmaceuticals-16-00993-f008:**
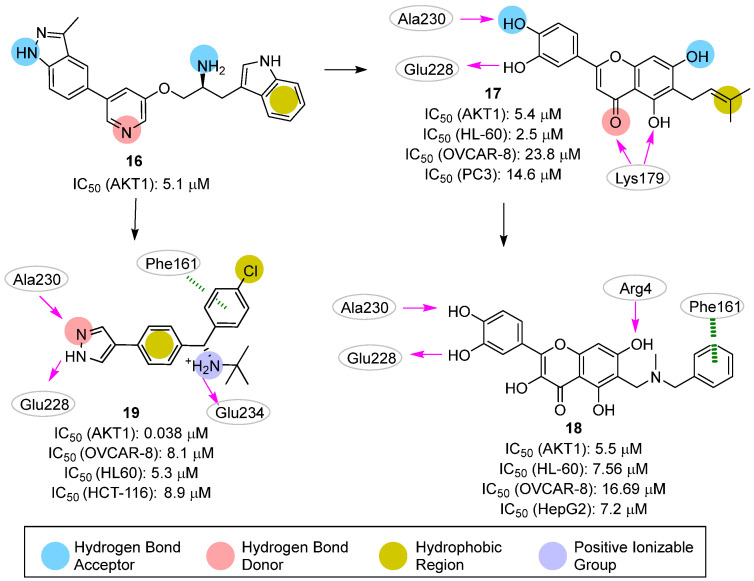
Targeting ATP-binding site. Known AKT1 inhibitor **16** and validated hits **17–19**. The predicted polar intermolecular interactions are illustrated as follows: magenta arrow, hydrogen bond; green dotted lines, π-π interaction. The original (**16** and **17**) [28] and the refined (**19**) [30] pharmacophore models are also illustrated.

**Figure 9 pharmaceuticals-16-00993-f009:**
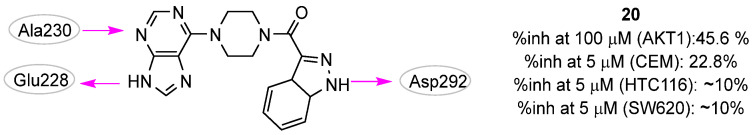
Targeting ATP-binding site. Compound **20** with the predicted polar intermolecular interactions illustrated as follows: magenta arrow, hydrogen bond.

**Figure 10 pharmaceuticals-16-00993-f010:**
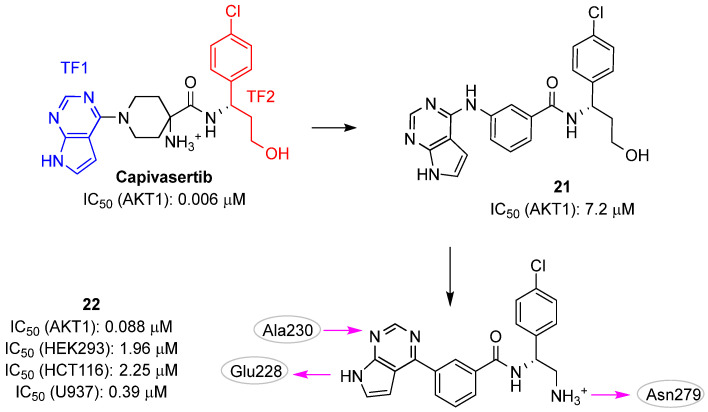
Targeting ATP-binding site. Known AKT1 inhibitor **capivasertib** and validated hits **21** and **22**. The predicted polar intermolecular interactions for compound **22** are illustrated as follows: magenta arrow, hydrogen bond. TF, terminal fragment.

**Figure 11 pharmaceuticals-16-00993-f011:**
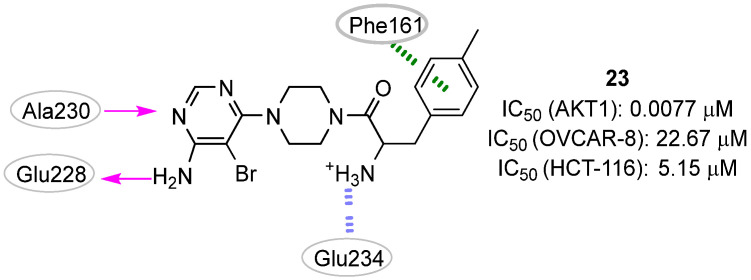
Targeting ATP-binding site. Validated hit **25** with the predicted polar intermolecular interactions illustrated as follows: magenta arrow, hydrogen bond; green dotted line, π-π interaction; purple dotted lines, π-cation interaction.

**Figure 12 pharmaceuticals-16-00993-f012:**
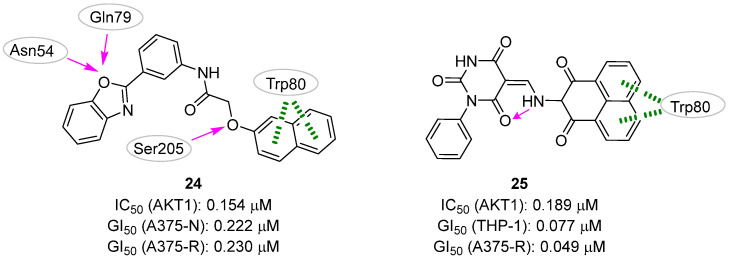
Targeting ATP-binding site. Validated AKT1 inhibitors **24** and **25** with the predicted polar intermolecular interactions for the latter compound illustrated as follows: magenta arrow, hydrogen bond; green dotted line, π-π interaction.

**Figure 13 pharmaceuticals-16-00993-f013:**
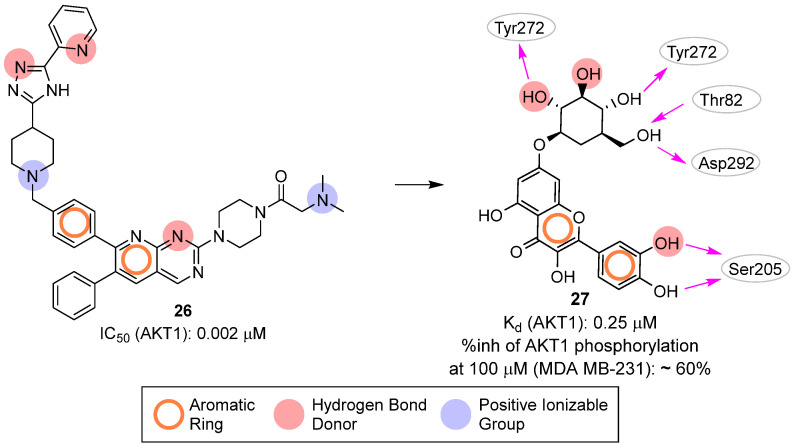
Targeting allosteric site. Known AKT1 inhibitor **26** and validated hit **27** and **24**. The predicted polar intermolecular interactions for the later compound are illustrated as follows: magenta arrow, hydrogen bond. The generated pharmacophore model is also highlighted.

**Figure 14 pharmaceuticals-16-00993-f014:**
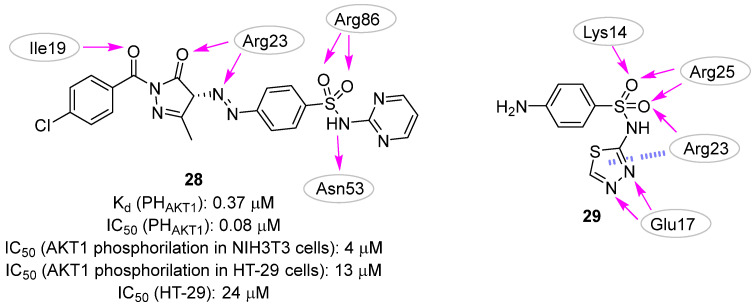
Targeting PI3P-binding site. Validated hits **28** and **29** with the predicted polar intermolecular interactions illustrated as follows: magenta arrow, hydrogen bond; blue dotted line, cation-π interaction.

**Figure 15 pharmaceuticals-16-00993-f015:**
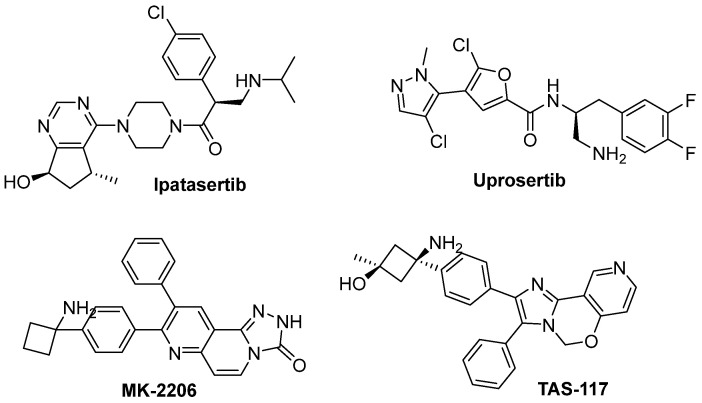
Chemical structure of the AKT clinical candidate inhibitors **Ipatasertib**, **Uprosertib**, **MK-2206**, and **TAS-117**.

**Table 1 pharmaceuticals-16-00993-t001:** Overview of collected AKT-targeted in silico studies according to the ligand binding site and the main computational methodology.

Site	Computational Methodology	Refs
ATP-binding site	docking-based	[20,22,23,24,25,26,27]
	pharmacophore-based	[28,29,30,31]
	ML	[32]
	QSAR	[33]
Allosteric site	docking-based	[34,35]
	pharmacophore-based	[36]
PI3P-binding site	pharmacophore-based	[37,38]

**Table 2 pharmaceuticals-16-00993-t002:** Summary of calculated and experimental parameters for compounds **29**–**31**.

Compound	LogP ^a^	Caco-2 ^a^	K_d_ (μM)	K_i_ (μM)	pAKT1 Inhibition (IC_50_, μM)	Cellular Growth Inhibition (IC_50_, μM)
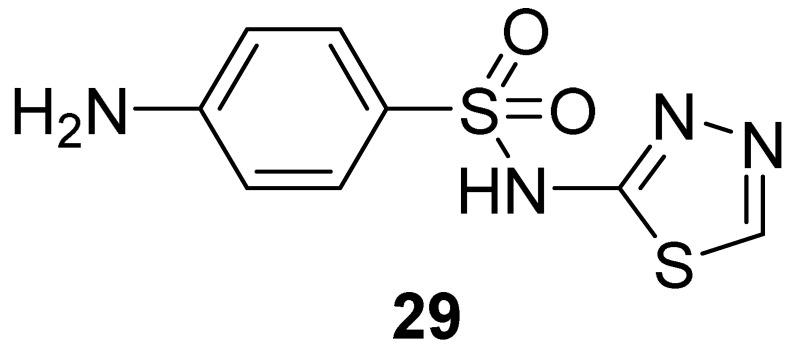	0.13	0.3	0.45 ± 0.1	>50.0	20 ^b^/25 ^c^	NI ^b^/NI ^c^
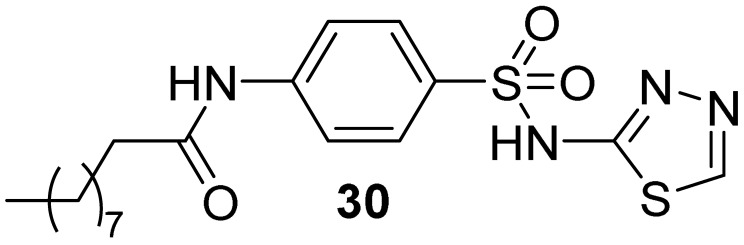	4.93	10.1	19.6 ± 4.9	21.8 ± 1.8	10 ^b^/15 ^c^	127 ^b^/90 ^c^
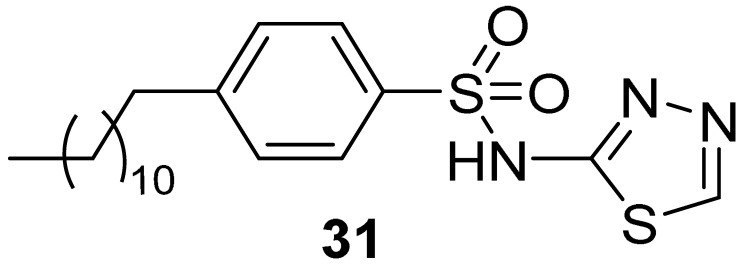	7.54	0.1	40.8 ± 2.5	2.4 ± 0.6	6 ^b^/10 ^c^	65 ^b^/30 ^c^

^a^ Predicted value; ^b^ Panc-1 (human pancreatic cancer cells); ^c^ MiaPaCa-2 (human pancreatic cancer cells).

**Table 3 pharmaceuticals-16-00993-t003:** Active clinical trial studies on AKT inhibitors for cancer therapy.

Compound	Figure	IC_50_	Ligand Binding Site	Clinical Phase	NCT Number
**Capivasertib** **(AZD5363)**	Figure 10	6 nM (AKT1)	ATP-binding site	Phase I	NCT01226316, NCT04556773
				Phase I/II	NCT01992952, NCT02208375, NCT03742102
				Phase II	NCT02117167, NCT02299999, NCT02465060, NCT02664935, NCT04439123
				Phase III	NCT03997123, NCT04305496
**Ipatasertib** **(GDC-0068)**	Figure 15	5 nM (AKT1)	ATP-binding site	Phase I	NCT03959891
				Phase I/II	NCT03280563, NCT03424005, NCT03853707
				Phase II	NCT02465060, NCT03395899, NCT03498521, NCT04464174, NCT04591431, NCT04632992, NCT05498896
				Phase III	NCT03072238, NCT04060862
**Uprosertib** **(GSK2141795)**	Figure 15	9.6 nM (AKT1)	ATP-binding site	Phase I/II	NCT01902173
**MK-2206**	Figure 15	8 nM (AKT1)	Allosteric site	Phase I	NCT01480154
				Phase II	NCT01251861, NCT01306045
**TAS-117**	Figure 15	0.55 nM (AKT1)	Allosteric site	Phase II	NCT04770246

**Table 4 pharmaceuticals-16-00993-t004:** List of available PDB structures for AKT1, AKT2 and AKT3.

Pocket	Domain	PDB ID	Release Date	Resolution (Å)	Exp. IC_50_ (nM)	Notes
**AKT1**						
no ligands	PH	1UNP	2004	1.65		
	PH	1UNR	2004	1.25		
	PH	2UZR	2007	1.94		
	Kinase	6BUU	2018	2.4		
	Kinase	6NPZ	2019	2.12		
	Full-length	7APJ	2021	2.05		Complexed with antibody
	PH	7MYX	2022	1.39		
ATP-binding site	Kinase	3CQU	2008	2.2	151	
	Kinase	3CQW	2008	2	318	
	Kinase	3MV5	2010	2.47	180	
	Kinase	3MVH	2010	2.01	0.5	
	Kinase	3OCB	2010	2.7	5	
	Kinase	3OW4	2010	2.6	22	
	Kinase	3QKK	2011	2.3	300	
	Kinase	3QKL	2011	1.9	9	
	Kinase	3QKM	2011	2.2	38	
	Kinase	4EKK	2012	2.8	*NR* ^a^	AMP-PNP
	Kinase	4EKL	2012	2	36.9	Clinical candidate for cancer (**Ipatasertib**)
	Kinase	4GV1	2013	1.49	4	Clinical candidate for cancer (**Capivasertib**)
	Kinase	6CCY	2018	2.18	3	
Allosteric site	Full-length	3O96	2010	2.7	58	
	Full-length	4EJN	2012	2.19	5	
	Full-length	5KCV	2016	2.7	8	Clinical candidate Proteus syndrome (**Miransertib**)
	Full-length	6HHF	2019	2.9	0.5	Covalent binder
	Full-length	6HHG	2019	2.3	9.1	Covalent binder
	Full-length	6HHH	2019	2.7	126	Covalent binder
	Full-length	6HHI	2019	2.7	3.6	Covalent binder
	Full-length	6HHJ	2019	2.3	3	Covalent binder
	Full-length	6S9W	2019	2.3	39	Covalent binder
	Full-length	6S9X	2019	2.6	381	Covalent binder
	Full-length	7NH4	2021	2.3	44	Covalent binder
	Full-length	7NH5	2021	1.9	112	Covalent binder
PI3P-binding site	PH	1H10	2003	1.4	*NR* ^a^	4IP
	PH	1UNQ	2004	0.98	*NR* ^a^	4IP
	PH	2UVM	2007	1.94	Ki = 80 nM	
	PH	2UZS	2007	2.46	*NR* ^a^	4IP, E17K mutation
**AKT2**						
no ligands	Kinase	1GZK	2003	2.3		
	Kinase	1GZN	2003	2.5		
	Kinase	1GZO	2003	2.75		
	Kinase	1MRV	2003	2.8		
	Kinase	1MRY	2003	2.8		
	PH	1P6S	2004	NMR		
ATP-binding site	Kinase	1O6K	2002	1.7	*NR* ^a^	ANP
	Kinase	1O6L	2002	1.6	*NR* ^a^	ANP
	Kinase	2JDO	2007	1.8	230	
	Kinase	2JDR	2007	2.3	0.5	
	Kinase	2UW9	2007	2.1	18	
	Kinase	2X39	2010	1.93	6	
	Kinase	2XH5	2010	2.72	27	
	Kinase	3D0E	2008	2	Ki: 4 nM	
	Kinase	3E87	2008	2.3	*NR* ^a^	
	Kinase	3E88	2008	2.5	0.6	
	Kinase	3E8D	2008	2.7	2	
**AKT3**						
no ligands	PH	2X18	2010	1.46		

^a^ NR: not reported.

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
