# Peer review of "Computer-Aided Identification of Kinase-Targeted Small Molecules for Cancer: A Review on AKT Protein"

_pharmaceuticals, 2023, doi:10.3390/ph16070993_

Round 1

Reviewer 1 Report

This is a very nice and concise review on AKT Protein and design of specific inhibitors

I would only ask for an illustration of the signaling pathway and clinical significance of AKT inhibitors (approved)

 Minor editing of English language required

Author Response

This is a very nice and concise review on AKT Protein and design of specific inhibitors. 

Q1) I would only ask for an illustration of the signaling pathway and clinical significance of AKT inhibitors (approved).

R1) We thank the reviewer for her/his comment. We added in Figure 2 a simplified illustration of the AKT pathway. Regarding the significance of the approved AKT inhibitors, unfortunately no AKT inhibitors reached the market yet. We added this concept in the introduction. 

Reviewer 2 Report

Primavera et al provided an informative and thorough review on the computational development of small molecule inhibitors for AKT proteins. The review covered inhibitors of three different pockets on AKT proteins and the inhibitors are categorized based on the method of identification.

This reviewer would recommend this paper for publication in Pharmaceuticals.

Author Response

We thank the reviewer for her/his positive comments.

Reviewer 3 Report

This reviewe is well-conceived and widely detailed, only sometimes difficutl to read.

Please, some minor revisions are suggested as follows:

- Figure 2, please revise it being quite confusing

- PDB codes should be accompanied by the proper citation along the whole manuscript, please check for this

- please, revise the last sections of the review in order to move Figure 15 and Tables 3-4 before of the the conclusion section

Author Response

This review is well-conceived and widely detailed, only sometimes difficult to read.

Please, some minor revisions are suggested as follows:

Q1) Figure 2, please revise it being quite confusing

R1) The figure 2 have been modified according to the reviewer's comment.

Q2) PDB codes should be accompanied by the proper citation along the whole manuscript, please check for this

R2) Done

Q3) please, revise the last sections of the review in order to move Figure 15 and Tables 3-4 before of the the conclusion section

R3) As general rule, we placed the figures and tables as near as possible to the corresponding citation in the main text. This is also specifically requested by the authors guidelines in which is stated "Schemes and Tables should be inserted into the main text close to their first citation". Following this guideline, we can  move Figure 15 and Table 3 before of the the conclusion section, but not Table 4.

Reviewer 4 Report

In their review, Primavera and co-workers presented a comprehensive overview of AKT inhibitors identified using computer-assisted drug design methodologies. The work is very well written and argued and I believe that it deserves to be published in Pharmaceuticals journal after addressing the following minor points:

1) At page 5, row 153, please change the sentence “the authors designed as series of derivatives (3-5, Figure 3) to mimic the interactions described as crucial between compound 2 and AKT2.” with the following one “the authors designed a series of derivatives (3-5, Figure 3) to mimic the interactions described as crucial between compound 2 and AKT2.”

2) Please, check throughout the manuscript that et al. is always italicized.

3) Please, check throughout the manuscript that the cited compounds are always indicated in bold.

4) Please, check that punctuation is always inserted after the reference.

5) At page 14, row 513, please modify the sentence “no activity decease was…” with the following one “no activity decrease was…”.

6) At page 17, row 617, please correct “The authors hypnotized that…” with “The authors hypothesized that…”

7) Please, check throughout the manuscript that in silico, in vitro and in vivo are always italicized.

Author Response

In their review, Primavera and co-workers presented a comprehensive overview of AKT inhibitors identified using computer-assisted drug design methodologies. The work is very well written and argued and I believe that it deserves to be published in Pharmaceuticals journal after addressing the following minor points:

Q1) At page 5, row 153, please change the sentence “the authors designed as series of derivatives (3-5, Figure 3) to mimic the interactions described as crucial between compound 2 and AKT2.” with the following one “the authors designed a series of derivatives (3-5, Figure 3) to mimic the interactions described as crucial between compound 2 and AKT2.”

R1) Done

Q2) Please, check throughout the manuscript that et al. is always italicized.

R2) Done

Q3) Please, check throughout the manuscript that the cited compounds are always indicated in bold.

R3) Done

Q4) Please, check that punctuation is always inserted after the reference.

R4) Done

Q5) At page 14, row 513, please modify the sentence “no activity decease was…” with the following one “no activity decrease was…”.

R5) Done

6) At page 17, row 617, please correct “The authors hypnotized that…” with “The authors hypothesized that…”

R6) Done

Q7) Please, check throughout the manuscript that in silico, in vitro and in vivo are always italicized.

R7) Done